# Adsorption Properties and Hemolytic Activity of Porous Aluminosilicates in a Simulated Body Fluid

**Olga Yu. Golubeva \*, Yulia A. Alikina, Elena Yu. Brazovskaya and Nadezhda M. Vasilenko**

Laboratory of Silicate Sorbents Chemistry, Institute of Silicate Chemistry of Russian Academy of Sciences, Adm. Makarova Emb., 2, 199034 St. Petersburg, Russia

**\*** Correspondence: olga_isc@mail.ru or golubeva@iscras.ru

**Abstract:** A study of the adsorption features of bovine serum albumin (BSA), sodium and potassium cations, and vitamin B1 by porous aluminosilicates with different structures in a medium simulating blood plasma was conducted. The objects of this study were synthetic silicates with a montmorillonite structure $Na_{2x}(Al_{2(1-x)},Mg_{2x})Si_4O_{10}(OH)_2 \cdot nH_2O$ (x = 0.5, 0.9, 1), aluminosilicates of the kaolinite subgroup $Al_2Si_2O_5(OH)_4$ with different particle morphologies (spherical, nanosponge, nanotubular, and platy), as well as framed silicates (Beta zeolite). An assessment of the possibility of using aluminosilicates as hemosorbents for extracorporeal blood purification was carried out. For this purpose, the sorption capacity of the samples both with respect to model medium molecular weight toxicants (BSA) and natural blood components—vitamins and alkaline cations—was investigated. The samples were also studied by X-ray diffraction, electron microscopy, and low-temperature nitrogen adsorption. The zeta potential of the sample's surfaces and the distribution of active centers on their surfaces by the method of adsorption of acid-base indicators were determined. A hemolytic test was used to determine the ability of the studied samples to damage the membranes of eukaryotic cells. Langmuir, Freundlich, and Temkin models were used to describe the experimental BSA adsorption isotherms. To process the kinetic data, pseudo-first-order and pseudo-second-order adsorption models were used. It was found that porous aluminosilicates have a high sorption capacity for medium molecular weight pathogens (up to 12 times that of activated charcoal for some samples) and low toxicity to blood cells. Based on the obtained results, conclusions were made about the prospects for the development of new selective non-toxic hemosorbents based on synthetic aluminosilicates with a given set of properties.

**Keywords:** aluminosilicates; kaolinite; montmorillonite; zeolites; hemosorbents; adsorption; albumin; vitamins; hemolytic activity; body fluid

## 1. Introduction

In recent years, more and more attention has been paid to studying the features of the adsorption of protein molecules from model solutions on the surface of sorbents of various natures, in particular, on the surface of clay minerals [1–6]. Interest in such research is related to the possibility of exploring the prospects for using clay minerals to remove proteins from wine, as well as the performance of membranes for protein separation, biosensors, or protein therapy platforms [2,7,8]. In addition, the relevance of these studies is associated with the need to develop biospecific sorbents for the selective adsorption of toxic substances of protein origin that accumulate in the body during oncological, immune, infectious, and other diseases [9,10].

Hemosorption is the most promising method of performing the sorption detoxification of the body [11–13]. Such sorption therapy is based on the adsorption ability of materials to remove toxic substances of various natures from the blood. The first and most common sorbents were materials based on activated carbon. Such materials are capable

of removing a variety of toxic molecules-exotoxins (poisons), cytokines, anti-inflammatory mediators, products of a bacterial nature, as well as those arising from cell breakdown [14–17]. However, activated charcoal-based materials have disadvantages, as in the process of hemosorption, there is a partial traumatization and death of blood cells. In addition, in the process of hemosorption on carbon sorbents, along with pathological components, a part of physiologically significant metabolites is removed. In this regard, a promising direction is the development of sorption technologies based on biospecific (selective) hemosorption.

There are a significant number of hemosorbents on the market, but none of them currently fully meet all the requirements for such materials, namely having a high sorption capacity with respect to toxins and metabolites, hemocompatibility, selectivity, and the ability to withstand certain sterilization methods without losing basic properties. The high sorption capacity of a number of inorganic adsorbents has great potential for medical use, however, according to some researchers, inorganic matrices, which usually mean natural porous minerals—clays, zeolites, etc.—are inferior to other adsorbents (activated carbon, synthetic and natural organic polymers) in terms of biocompatibility [18–20]. This problem can be solved by using synthetic inorganic matrices with the following desired characteristics: high sorption characteristics and hemocompatibility due to the absence of impurity phases, controlled chemical and dispersion composition, as well as a certain particle morphology and specified porosity in a wide range (from nano- to macro- and mesopores), which allows the adsorption of biological molecules of different sizes.

Medical sorbents must meet certain requirements—a high degree of chemical purity, a minimum content of impurities, a smooth surface relief, a high sorption capacity for removed substances, and the presence of hemocompatibility [21,22]. Under the conditions of directed hydrothermal synthesis, the porous aluminosilicates of various structures can be obtained with specified characteristics, such as a certain phase and chemical composition, given particle size and morphology, as well as porous textural and sorption characteristics. Preliminary studies of the cytotoxicity and hemolytic activity of synthetic samples of aluminosilicates showed that they do not have the toxicity that is characteristic of natural minerals, which indicates that it is promising to study the possibility of their use as medical sorbents [23,24].

The present work presents the results of a study of porous textural properties, surface properties, hemolytic activity, as well as the features of adsorption by synthetic porous aluminosilicate sorbents with different porosities and particle morphologies from a medium simulating blood plasma, bovine serum albumin, sodium and potassium cations, as well as vitamin B1. Framework aluminosilicates (zeolites), layered silicates with montmorillonite structure, as well as layered silicates of the kaolinite subgroup with spherical, sponge, and platy morphologies were selected as objects for this study.

Bovine serum albumin (BSA) is a water-soluble globular protein (with an approximate molecular size of 9 nm x 8 nm x 6 nm.) [25,26], which is part of the blood serum and blood cytoplasm of animals and plants. Albumin refers to proteins with an average molecular weight of 67–69 kDa. BSA is often used to understand the adsorption mechanism of proteins at solid/liquid interfaces. In this study, BSA acts as a marker of medium molecular weight proteins. It is known that pathogenic compounds formed in the body during oncological, immune, infectious, and other diseases belong to proteins of medium molecular weight [1,27]. The adsorption of albumin by clay minerals has been widely studied [1–6,28], especially with regard to biosensors. Since the value of the isoelectric point of albumin is 5, most studies were carried out with solutions having acidic pH values (4.5) and sometimes at elevated temperatures. At the same time, the requirements for hemosorbents impose certain requirements on the experiments being carried out—the pH values must correspond to the pH of the blood plasma (neutral) and the temperature of the study should not exceed 37 °C. To replicate the conditions of hemosorption as accurately as possible, in this work, studies of the adsorption process were carried out at room temperature at neutral pH, from the medium of a synthetic biological fluid, prepared in

accordance with the chemical analysis of human body fluids, with ion concentrations nearly equal to those of the inorganic components of human blood plasma [29,30]. Alkaline cations and vitamins, the adsorption of which was also considered in this work, are essential microelements that are part of the blood and affect the state of the cardiovascular and other human systems. Sodium, potassium, calcium, and magnesium play a central role in the normal regulation of blood pressure [31]. A marked reduction in sodium and potassium intake is effective, even in treating severe hypertension. Thiamin, or vitamin B1, is an essential water-soluble vitamin that acts as a coenzyme in carbohydrate and branched-chain amino acid metabolism.[32] Therefore, the loss of mineral substances during the process of hardware blood purification in the process of hemosorption is extremely undesirable.

The results of the study of the adsorption capacity of synthetic aluminosilicate samples with different porosity (for example, the maximum diameter of zeolite cavities does not exceed 1 nm; and montmorillonites have the ability to change the interlayer distance over a wide range—from 1 Å to complete exfoliation into individual layers) and with different surface properties, this will allow us to evaluate the possibility of developing universal and selective sorbents for carrying out the adsorption of substances of different molecular weights and different molecular sizes.

## 2. Materials and Methods

### 2.1. Reagents

The following reagents were used for the synthesis and analysis of the samples: tetraethoxysilane TEOS ($(C_2H_5O)_4Si$, special purity grade, ≥99.0%), aluminum nitrate $Al(NO_3)_3 \cdot 9H_2O$ (reagent grade, ≥97.0%), magnesium nitrate $Mg(NO_3)_2 \cdot 6H_2O$ (reagent grade), nitric acid $HNO_3$ (reagent grade, 65 wt%), aqueous ammonia (25 wt% $NH_3$), ethanol $C_2H_5OH$ (96 wt%), hydrochloric acid HCl (35–38 wt%), sodium hydroxide solution (50 wt% in water), raw halloysite nanotubes (Sigma-Aldrich, Product of Applied Minerals, USA), potassium hydroxide (KOH, 45% aqueous solution), silica sol (LUDOX HS_40, 40%), aluminum sulfate ($Al_2(SO_4)_3 \cdot 18H_2O$, ≥98%), tetraethylammonium hydroxide ($(C_2H_5)_4NOH$, 35% aqueous solution, Sigma), activated charcoal (MW 12.01 g/mol, Fluka Analytical), bovine serum albumin (lyophilized pH~7, Biowest), and vitamin B1 (Thiamine hydrochloride, reagent grade ≥ 99%, Hubei Maxpharm Industries).

Simulated body fluid (SBF) was prepared according to the procedure in [25] using the following reagents: NaCl (98%, NevaReactiv), $NaHCO_3$ (99.5%), KCl (NevaReactiv, 99%), $Na_2HPO_4 \cdot 2H_2O$ (98%, Chimmed), $MgCl_2 \cdot 6H_2O$ (98%, NevaReactiv), $CaCl_2 \cdot 2H_2O$ (98%, NavaReactiv), and $(CH_2OH)_3CNH_2$ (Trizma base, Sigma, MW127.14 g/mol). Solutions were prepared in deionized water (Vodolei, NPP Khimelektronika, Russia) with a specific conductivity no higher than 0.2 $\mu$S/cm.

### 2.2. Synthesis of Aluminosilicates

Porous aluminosilicates of various structural types and with different particle morphologies were chosen as the objects of this study. All studied aluminosilicates were synthetic, with the exception of halloysite nanotubes. The main characteristics of the samples, their chemical formulas, and structural types are given in Table 1.

Samples with a montmorillonite structure corresponding to the ideal chemical formula $Na_{2x}(Al_{2(1-x)},Mg_{2x})Si_4O_{10}(OH)_2 \cdot nH_2O$ with various degrees of isomorphic substitution magnesium atoms in octahedral layers were chosen as objects of this study: with x = 1 ($Mg_3Si_4O_{10}(OH)_2 \cdot nH_2O$), x = 0.9 ($Na_{1.8}Al_{0.2}Mg_{1.8}Si_4O_{10}(OH)_2 \cdot H_2O$), and x = 0.5 ($Na_{1.0}Al_{1.0}Mg_{1.0}Si_4O_{10}(OH)_2 \cdot nH_2O$). Samples corresponding to the $Al_2Si_2O_5(OH)_4$ kaolinite formula were synthesized under conditions that made it possible to obtain various particle morphologies—spherical, sponge, and platy. In addition, the sorption and physicochemical properties of the samples were compared with the results of a study of natural

halloysite $Al_2Si_2O_5(OH)_4 \cdot nH_2O$ with nanotubular morphology. The zeolite of the structural type Beta was also studied as an object of this study.

The synthesis of all samples was carried out under hydrothermal conditions according to previously developed methods [23,33–37]. The resulting product was washed with water and dried. For zeolite samples, an additional decationization procedure was carried out, that is, the removal of alkaline cations $K^+$ and $Na^+$ localized in large cavities. Sample decationization was carried out by the triple treatment of zeolites with an ammonium salt solution followed by drying at 120 °C and the decomposition of the ammonium ion $NH^{4+}$ at 600 °C for 1 h. In addition, the initial zeolite was preliminarily calcined for 2 h at a temperature of 350 °C in order to remove the adsorbed water and residues of organic molecules (tetraethylammonium) from the pores of the zeolite.

**Table 1.** Main characteristics of the studied samples.

| Samples Designation | Mineralogical Name | Structural Type | Chemical Formula (by Synthesis) | Particles Morphology | Synthesis Conditions | | Content, wt% | | | | |
|---|---|---|---|---|---|---|---|---|---|---|---|
| | | | | | T, °C | t, h | SiO₂ | Al₂O₃ | MgO | Loss on Ignition, % | Additionally. |
| MT-Al0 | Montmorillonite | LS | $Mg_3Si_4O_{10}(OH)_2 \cdot nH_2O$ | layers | 250 | 72 | 59.39 | 0 | 28.63 | 11.45 | - |
| MT-Al0.2 | Montmorillonite | LS | $Na_{1.8}Al_{0.2}Mg_{1.8}Si_4O_{10}(OH)_2 \cdot H_2O$ | layers | 350 | 72 | 58.10 | 5.32 | 18.31 | 14.75 | Na₂O 3.52 |
| MT-Al1.0 | Montmorillonite | LS | $Na_{1.0}Al_{1.0}Mg_{1.0}Si_4O_{10}(OH)_2 \cdot nH_2O$ | layers | 350 | 72 | 53.00 | 22.82 | 8.04 | 13.45 | Na₂O 2.69 |
| Kaol-sph | Kaolinite | LS | $Al_2Si_2O_5(OH)_4$ | spheres | 220 | 72 | 44.74 | 37.22 | 0 | 14.74 | - |
| Kaol-sponge | Kaolinite | LS | $Al_2Si_2O_5(OH)_4$ | nanosponges | 220 | 72 | 43.77 | 36.14 | 0 | 15.79 | - |
| Kaol-pl | Kaolinite | LS | $Al_2Si_2O_5(OH)_4$ | plates | 350 | 96 | 45.84 | 39.48 | 0 | 14.05 | - |
| Hal | Halloysite | LS | $Al_2Si_2O_5(OH)_4 \cdot nH_2O$ | nanotubes | - | - | 46.22 | 36.38 | 0 | 16.04 | - |
| Beta | Zeolite Beta | FS | $H^+_7[Al_7Si_{57}O_{128}] \cdot nH_2O$ | spheres | 135 | 48 | 69.38 | 8.54 | 0 | 20.09 | Na₂O 0.3, K₂O 0.2 |

Designations: LS—layered silicate; FS—framed silicate.

### 2.3. Characterization

The X-ray phase analysis of the samples was carried out using a powder diffractometer Rigaku Corporation, SmartLab 3 (CuKα-radiation, operating mode-40 kV/40 mA; semiconductor point detector (0D)-linear (1D), θ-θ geometry, measurement range $2\vartheta = 5$–$70°$ (step $2\theta = 0.01°$), speed 5°/min).

The samples were chemically analyzed to gravimetrically determine the Si, Mg, and Al contents using a quinolate of the silicon molybdenum complex and by complexometric titration. The sodium and potassium content of the studied samples was determined by atomic absorption spectroscopy (Thermo scientific iCE 3000, Waltham, MA, USA).

The textural parameters of the materials were determined by means of the low-temperature adsorption–desorption of nitrogen. The isotherms were collected using a Quantachrome NOVA 1200e instrument (Quantachrome Instruments, Boynton Beach, FL, USA). Degassing was performed at 300 °C for 12 h. The specific surface area of the sample was calculated by the BET method [38] using NOVAWin (USA) software. The pore size distribution and mean pore diameter were calculated by the Barret-Joyner-Halenda (BJH) method from the desorption curve [39].

The morphology of the samples was studied by scanning electron microscopy (SEM) by using a Carl Zeiss Merlin instrument (Oberkochen, Germany) with a field emission cathode. The beam current and accelerating voltage were 2 nA and 21 kV, respectively. The device was equipped with a two-beam workstation with focused ion and scanning electron beams, a Carl Zeiss Auriga laser with a field emission cathode, a GEMINI electron optics column, and an oil-free vacuum system with a beam current range of 400 pA and an acceleration voltage of 1.5–4 kV. The powders of the samples were directly planted on conductive carbon tape without additional processing.

The electrokinetic (zeta) potential of the samples was determined using the particle size and zeta potential analyzer NaniBrook 90 PlusZeta (Brookehaven Instruments

Corporation, USA). The samples were a suspension obtained by dispersing 50 mg of sample in 20 mL of deionized water. Before the measurements, the suspension was subjected to low power (50 W) ultrasonication for two minutes on an ultrasonic processor UP50H.

The functional composition of the surface of the samples was studied by the method of the adsorption of acid-base indicators with different pKa values in the range from −4.4 to 14.2, undergoing a selective adsorption on the surface of active centers with the corresponding pKa values according to the procedure described in [40]. The content of adsorption centers was determined from the change in the optical density of the aqueous solutions of indicators using UV absorption spectroscopy (LEKISS2109UV spectrophotometer).

The adsorption properties of the samples with respect to BSA were studied under static conditions from BSA solutions in SBF with an albumin concentration of 2.4 g/L. The experiments were carried out at room temperature (25 ± 1 °C), which corresponds to the conditions of the hemosorption procedure. To a weighed portion of the sorbent (30 mg), 10 mL of a BSA solution in SBF was added, and the mixture was stirred on a magnetic stirrer for the time necessary to plot the kinetic curves (from 1 to 30 h). After the experiment was completed, the sample was centrifuged. The protein concentration in the supernatant were analyzed with a UV–Vis spectrophotometer (SHIMADZU UV-2600/2700) at 278 nm. Each point of the kinetic curve was taken as the average of three measurements. The BSA concentration was determined using UV–Vis absorption spectroscopy (Shimadzu UV-2600/2700, Shimadzu Europa GmbH) by the optical density at a wavelength of 278 nm.

The capacity of the sorbent, mg/g (the amount of adsorbed substance), was determined by the following Formula (1):

$$X = (C_i - C_f) \, V_s / \, m_s, \tag{1}$$

where $C_i$ is the initial concentration of albumin solution, g/L; $C_f$ is the final concentration after sorption, g/L; $V_s$ is the volume of albumin solution, L; and $m_s$ is the weight of the sorbent sample, g.

To process the kinetic data, pseudo-first-order (PFO) and pseudo-second-order (PSO) adsorption models [41,42] were used. The kinetic expression for PFO, based on the capacitance of a solid, is written in the following form:

$$q_t = q_e(1 - e^{-k_1 t}), \tag{2}$$

where $q_t$ and $q_e$ are the sorption capacity at time t and in equilibrium (mg/g), and $k_1$ is the PFO reaction rate constant, min$^{-1}$

The mathematical expression for the PSO kinetic model is as follows:

$$q_t = \frac{q_e^2 k_2 t}{1 + q_e k_2 t} \tag{3}$$

where $q_t$ and $q_e$ are the sorption capacity at time t and in the equilibrium state (mg/g) and $k_2$ is the PSO rate constant (g/(mg·min)).

The study of the equilibrium adsorption of BSA was carried out at an initial albumin concentration in the range from 100 to 2400 mg/L. For this, 32 mg of a sorbent sample with a weighing accuracy of ±0.0002 g was dispersed in 10 mL of BSA solution in SBF with a given concentration. The experiments were carried out in a static mode in closed glass bottles with a volume of 20 mL with stirring for the time necessary to achieve adsorption equilibrium (from 4 to 24 h depending on the structural type of aluminosilicates). The samples were filtered and the albumin concentration in the filtrate was determined as the arithmetic mean of three measurements. To establish the patterns of sorption, the equations of isotherms were calculated according to the most widely used Langmuir, Freundlich, and Temkin models [43–45]. The parameters of the adsorption equations were calculated by the method of nonlinear regression using the OriginPro 8 program.

The adsorption capacity of the samples for vitamin B1 was determined under static conditions at room temperature (25 ± 1 °C). Vitamin B1 solution in SBF (100 mg/L, 10 mL) was added to 30 mg of the sorbent and stirred on a magnetic stirrer for 1 h. After the experiment was completed, the sample was centrifuged. The vitamin concentration in the supernatant was analyzed with a UV–Vis spectrophotometer (Shimadzu UV-2600/2700, Shimadzu Europa GmbH) at 242 nm. Each concentration was taken as the average of three measurements.

The content of the sodium and potassium cations (in mmol/L) in the SBF solutions after contact with sorbent samples for 1 h was determined by atomic absorption spectroscopy (Thermo scientific iCE 3000, USA). To a weighed portion of the sorbent (30 mg), added 20 mL of SBF was and stirred on a magnetic stirrer for 1 h at room temperature. After the experiment was completed, the sample was centrifuged. The content of the cations in the initial SBF solutions corresponded to the reference values of the content of sodium and potassium cations in human blood plasma and amounted to 142 and 3.43 mmol/L, respectively. The sorption capacity of the samples (C, mg/g) was determined using the following formula (1).

A hemolytic test was used to determine the ability of the studied samples to damage the membranes of eukaryotic cells [46]. Human erythrocytes obtained from the peripheral blood of healthy donors by standard procedure [46,47] were used to determine the hemolytic activity. The studies were carried out according to the previously described method. [23,34]. The final concentration of aluminosilicate preparations in the incubated samples was 10 mg/mL and 0.3 mg/mL. The result of the study was presented as a percentage of hemolysis corresponding to the content of hemoglobin released from destroyed erythrocytes after the incubation of a suspension of erythrocytes with the studied samples of aluminosilicate.

## 3. Results and Discussion

The X-ray diffraction patterns of samples are shown in Figure 1. The comparison of the diffraction patterns of the samples with the bar charts of the standards allows us to conclude that the single-phase samples of specified structures, montmorillonite, kaolinite, halloysite, and Beta zeolite are used as initial samples. The results of the chemical analysis of the samples (Table 1) confirm that the samples studied are hydrous aluminosilicates with different Si/Al ratios.

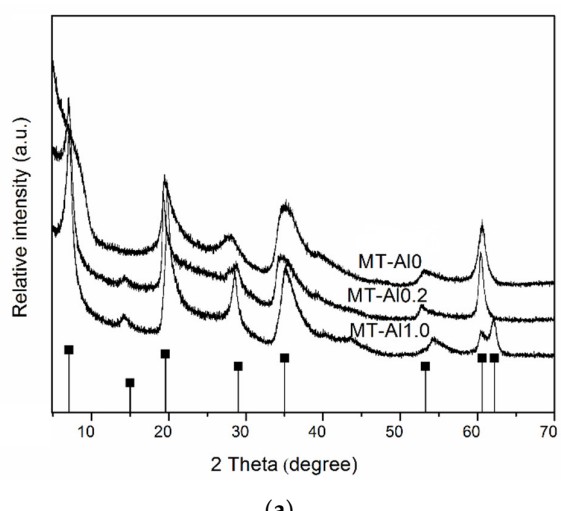

(a)

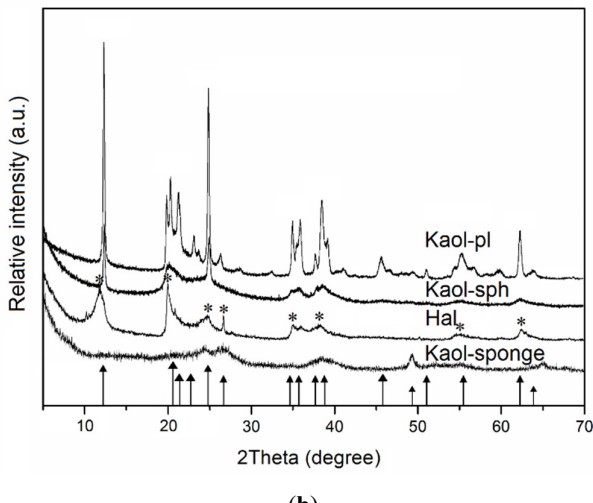

(b)

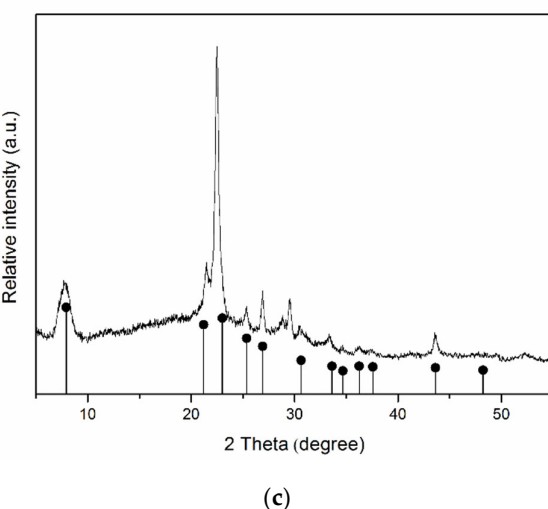

(**c**)

**Figure 1.** X-ray diffractions patterns of the samples: (**a**) aluminosilicates with montmorillonite structure; (**b**) aluminosilicates with kaolinite structure; and (**c**) zeolite Beta. Bar chart of the standards: ■—raw montmorillonite (PDF No. 48-74); ▲—raw kaolinite (PDF No. 79-1593); *—raw halloysite (PDF No 00–009-0453); ●—Beta (PDF No. 12-204).

Figure 2 shows the SEM images of the samples. It is observed that the aluminosilicate samples are characterized by different particle morphologies. Thus, the main morphology of the samples with the montmorillonite structure are layers self-organized into larger micron size agglomerates (Figure 2a–c). According to previous studies [33], the average particle size of montmorillonite is approximately 20 nm. Samples with a kaolinite structure were obtained with spherical, platy, and sponge morphologies. Particles with a spherical morphology have an average diameter of approximately 200–300 nm (Figure 2g). Samples with a nanosponge morphology are formed by aluminosilicate layers with a thickness of approximately 24–27 nm which are combined into micron-sized agglomerates (Figure 2f). Platy particles have a thickness of approximately 100 nm and an average lateral size of approximately 1 µm (Figure 2d). The raw halloysite has a nanotubular particle shape (Figure 2e). The nanotubes are approximately 700 nm long and 60 nm in diameter. Zeolite Beta particles have a spherical morphology with an average diameter of 300 nm (Figure 2h).

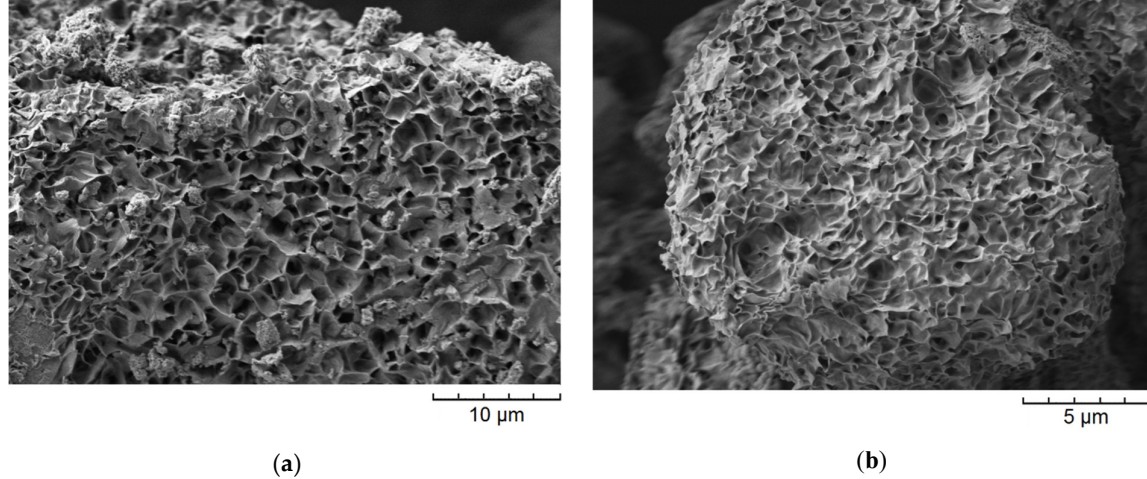

(**a**)　　　　　　　　　　　　　　　　　　　　(**b**)

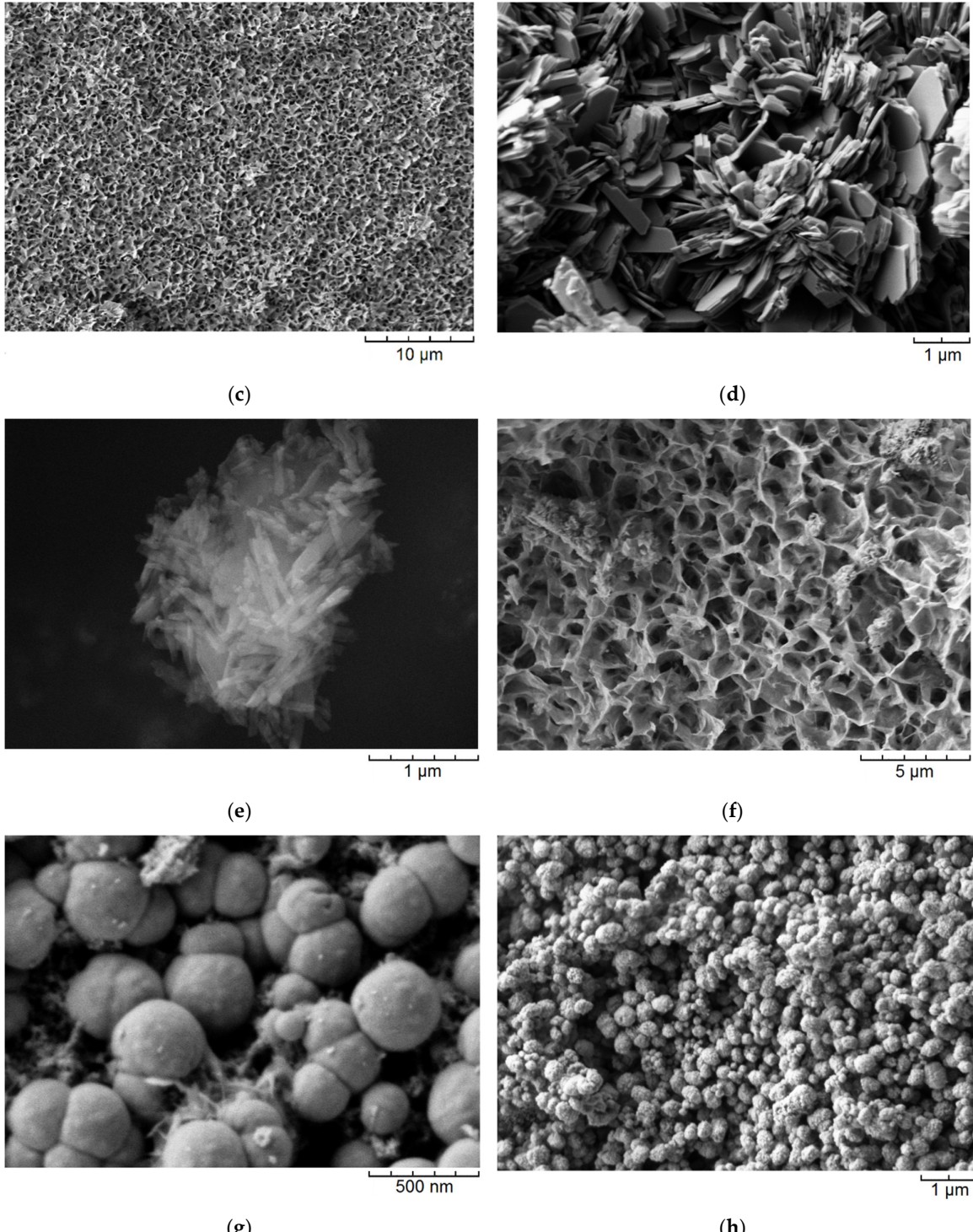

**Figure 2.** SEM images of the samples: (**a**)—MT-Al0; (**b**)—MT-Al0.2; (**c**)—MT-Al1.0; (**d**)—Kaol-pl; (**e**)—Hal; (**f**)—Kaol-sponge; (**g**)—Kaol-sph; and (**h**)—Beta. Samples are designated in accordance with the designations presented in Table 1.

Figure 3 shows low-temperature nitrogen adsorption curves for the studied aluminosilicate samples. All curves can be attributed to a type IV isotherm according to the IUPAC classification. This type of isotherm indicates the presence of both micro- and mesopores [48]. The hysteresis loops are of different shapes, which is associated with the

different types and shapes of pores in the samples. The shape of the hysteresis curves for kaolinite samples with a spherical and sponge morphology of particles, tubular halloysite, samples of montmorillonite, and Beta zeolite can be attributed to the *H2* type. This shape of the hysteresis loop points to complicated partly constricted pore network [49]. The shape of the hysteresis loop of the kaolinite sample with a platy morphology can be attributed to the *H3* type and indicates the presence of the aggregates of platy particles that form slit-like pores. The samples differ considerably in their specific surface area (SSA), which increases from 20 to 676 m²/g depending on the particle morphology (see Table 2). In addition, the samples differ in the average pore diameter. For example, an average pore diameter of MT-Al0 sample is 4.4 nm, while that of MT-Al0.2 and MT-Al1.0 montmorillonites is 1.8 and 3.8 nm, respectively. The average pore size of the samples with the structure of kaolinite and zeolite Beta is 3.7 nm. For the Beta zeolite, this value most likely characterizes the secondary porosity, since the average size of the channels and cavities of this zeolite does not exceed 0.8 nm [50].

Along with the porous textural properties of the sorbents, an important role in the choice of sorption materials in medicine is played by the chemical nature of their surface, namely the composition and number of functional groups on the surface. The chemistry of surface compounds determines the course of donor–acceptor interactions, which significantly affects the spectrum of absorbed molecules, and consequently, biochemical parameters ([51]).

The distribution of the adsorption sites on the surface of the studied samples as a function of their $pK_a$ values is shown in Figure 4. These results indicate the presence of different types of adsorption centers on the surface including a Lewis base ($pK_a \leq 0$, formed by oxygen atoms) and acidic ($pK_a \geq 14$, formed by silicon atoms), Bronsted acidic ($0 < pK_a < 6$), neutral ($pK_a \sim 6–8$), and basic ($8 < pK_a < 14$) sites corresponding to hydroxyl groups [40].

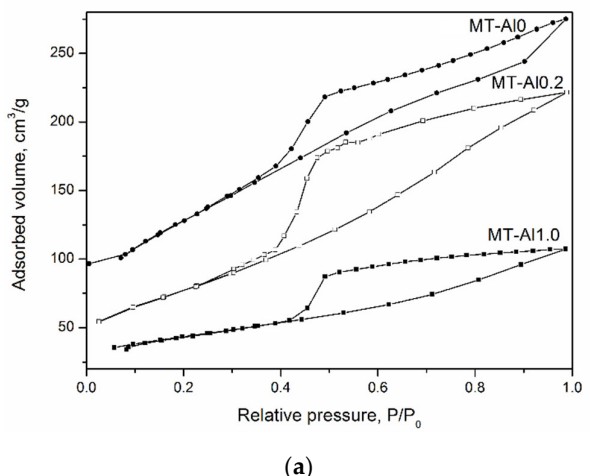

(**a**)

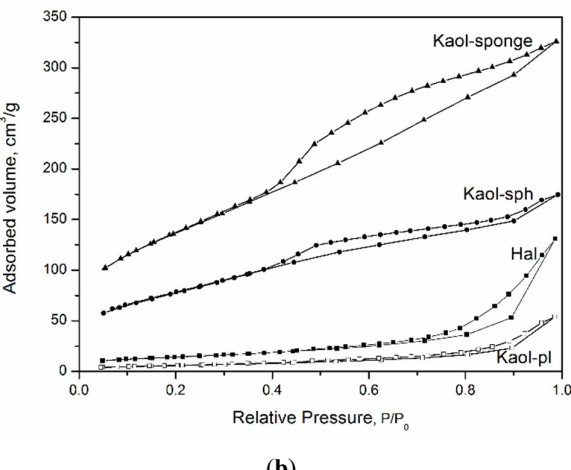

(**b**)

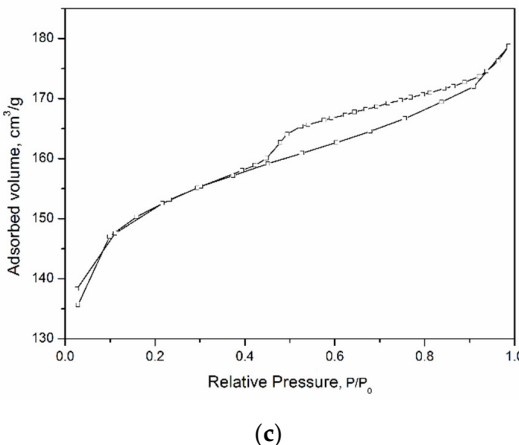

(**c**)

**Figure 3.** $N_2$ adsorption–desorption isotherms of the samples: (**a**)—aluminosilicates with montmorillonite structure; (**b**)—aluminosilicates with kaolinite structure; and (**c**)—zeolite Beta.

**Table 2.** Properties of the studied samples of aluminosilicates.

| Samples | SSA [a], m²/g | ζ (pH 7), mV | Sorption Capacity for Cations, mg/g | | Sorption Capacity for Vitamin B₁, | Hemolysis%, at Sample Concentration | |
|---|---|---|---|---|---|---|---|
| | | | Na⁺ | K⁺ | mg/g | 10 mg/mL | 0.3 mg/mL |
| MT-Al0 | 549 | −15.1 ± 0.2 | 0 | 0 | 22.4 ± 0.9 | 5.1 ± 0.9 | 0.6 ± 0.5 |
| MT-Al0.2 | 320 | −33.3 ± 0.3 | 3.3 ± 0.9 | 0.20 ± 0.08 | 39.9 ± 0.1 | 58.5 ± 5.9 | 2.0 ± 1.2 |
| MT-Al1.0 | 190 | −34.1 ± 0.9 | 4.2 ± 0.5 | 0.14 ± 003 | 31.6 ± 0.8 | 86.9 ± 9.0 | 14.9 ± 4.6 |
| Kaol-sph | 240 | −18 ± 0.8 | 2.7 ± 0.3 | 0.13 ± 0.02 | 1.37 ± 0.1 | 23.1 ± 2.6 | 3.9 ± 4.6 |
| Kaol-sponge | 470 | −20 ± 0.6 | 0 | 0 | 23.3 ± 0.2 | 27.0 ± 7.0 | 2.3 ± 0.8 |
| Kaol-pl | 22 | −19 ± 0.9 | 3.1 ± 0.3 | 0.12 ± 0.03 | 0.8 ± 0.2 | 66.1 ± 1.8 | 3.8 ± 1.6 |
| Hal | 41 | −28 ± 0.4 | 4.3 ± 0.2 | 0.13 ± 0.03 | 1.1 ± 0.1 | 97.5 ± 8.6 | 25.5 ± 9.5 |
| Beta | 676 | −32.2 ± 0.6 | 9.1 ± 0.6 | 0.19 ± 0.05 | 29.5 ± 0.7 | 15.7 ± 1.7 | 0.9 ± 0.2 |
| Carbon | 360 | | 0.6 ± 0.4 | 0 | 6.7 ± 0.9 | 8.7 ± 2.3 | 0.9 ± 0.2 |

Designations: SSA—specific surface area (m²/g); ζ (pH 7)—zeta potential of the surface as pH 7, mV, [a]—relative error in the specific surface area (SSA) value is 1%.

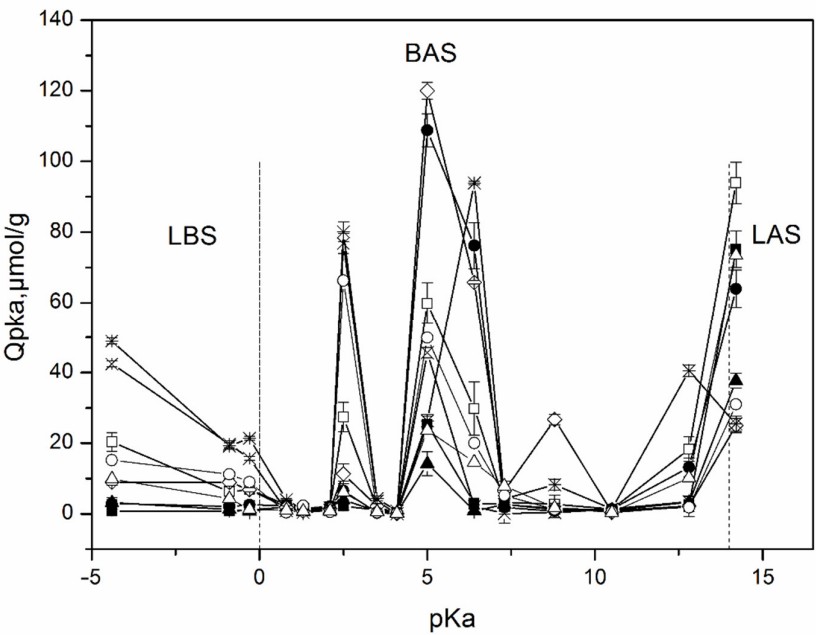

**Figure 4.** Distribution of the adsorption centers as a function of their pKa values on the surface of aluminosilicate samples and activated carbon: ◊—MT-Al0; ○—MT-Al0.2; △—MT-Al1.0; ●—Kaol-sph; ▲—Kaol-sponge; □—Kaol-pl; ■—Hal; ×—Beta; ∗—activated carbon. LBS—Lewis basic sites; BAS—Brønsted acidic sites; LAS—Lewis acidic sites.

An analysis of the surface of the studied samples by the method of adsorption of acid-base indicators allows us to draw conclusions about the distribution of active centers on the surface, as well as about the change in the strength and ratio between these centers with a change in the chemical composition of aluminosilicates and their morphology. Samples of all compositions contain weakly acidic Brønsted acid sites (BASs) with pK$_a$ 5 and BAS with increased acidity with pK$_a$ 2.5. At the same time, the maximum amount of BAS with pK$_a$ 5 is typical for a sample of MT-Al0 and kaolinite with a spherical particle morphology. Their smallest amount is observed for a kaolinite sample with nanosponge morphology. All samples are characterized by the presence of Lewis basic sites (LBSs) with pK$_a$-4.4. At the same time, the number of such sites is at its maximum for the zeolite sample, and somewhat less for the kaolinite samples with a platy morphology. In other samples, the content of LBSs with pK$_a$ 4.4 is quite low. The kaolinite sample with platy particle morphology is also characterized by a significant content of LAS with pK$_a$ 14.2. The activated carbon sample is characterized by a high content of active sites with pK$_a$ 6.4, corresponding to Brønsted neutral centers, and a rather low number of active sites with pK$_a$ 5. Most aluminosilicate samples, on the contrary, are characterized by a high number of active centers with pK$_a$ 5 and a small number, or even the complete absence, of active sites with pK$_a$ 6.4. An exception is a sample of kaolinite with a spherical morphology of particles and MT-Al0, which have a large number of active centers with pK$_a$ 5 and with pK$_a$ 6.4. The MT-Al0 sample is also characterized by a large number of active centers with pK$_a$ 8.8. The rest of the samples have practically no active centers in this region.

Comparison of the obtained data with the results of the chemical analysis of the samples and the study of the morphology of their particles allows us to conclude that both the chemical composition and morphology affect the distribution of active centers on the surface of silicate sorbents. Thus, the MT-Al0 sample studied in this work, which does not contain aluminum in its composition, but contains magnesium oxide, is characterized by the largest number of BAS among all samples with pK$_a$ 8.8 and 2.5. Samples of aluminosilicates of the same chemical composition, but with different particle morphologies, such

as kaolinite with spherical, platy, and sponge morphologies, as well as nanotubular halloysite, have a different functional composition of active centers on their surface, which may be due to the different availability of these centers, as determined by their morphology. Thus, the largest amount of BAS with $pK_a$ 5 in this series of samples is characteristic of a sample with a spherical particle shape. The sample with a platy morphology has the highest amount of LAS with $pK_a$ 14.2. The sample with nanotubular morphology as a whole has the smallest number of active centers, however, the amount of LAS with $pK_a$ 14.2 in this sample is significant. A comparison of the functional composition of the surface of the studied samples with activated carbon shows that silicates have more active centers both in terms of their number and in terms of their diversity.

The results of the study of the zeta potential of the surface of the samples are presented in Table 2. All the studied samples have a negative surface zeta potential at pH 7, which is typical for aluminosilicates, and ranges from −25 ± 8 mV. Somewhat more negative values are typical for the samples of montmorillonite and zeolite (approximately −30 mV) than for samples of kaolinite—from −18 to −28 mV—depending on the particle morphology. The least negative surface charge among all the samples is characteristic of the MT-Al0 sample (−15 mV), which is associated with the absence of isomorphic substitutions in the octahedral magnesium–oxygen layers. The magnitude of the surface charge can make a significant contribution to the nature of the adsorption of charged molecules. Thus, aluminosilicates, having a negative surface charge, usually sorb positively charged ions from aqueous solutions very well (e.g., methylene blue, vitamin B1) [23,52], and sorb negatively charged ions (e.g., azorubine, 5-fluoracil) [23,53] to a much lesser extent. It is known that the albumin molecule has a net negative charge at a physiological pH [54], which can lead to difficulties in the adsorption of albumin by porous aluminosilicates.

Figure 5 shows the kinetic curves of albumin adsorption by the studied samples of aluminosilicates. The data obtained allow us to conclude that the time to achieve sorption equilibrium, depending on the sorbent, varies in the range from 4 to 24 h. The shortest time to achieve sorption equilibrium (1 and 2 h) is characteristic for activated carbon and Beta zeolite, respectively. It can be seen that the samples of montmorillonite have the highest sorption capacity with respect to albumin, both with isomorphic substitutions (MT-Al1.0 and MT-Al0.2) and without them (Sap), however, the time to reach adsorption equilibrium for them is the longest and reaches 20–24 h. Upon contact with albumin for 24 h, the sorption capacity of the montmorillonite samples reaches 220–250 mg/g. Montmorillonite MT-Al1.0 has the highest sorption capacity. For 24 h of contact, the sorption capacity of MT-Al1.0 reaches 256 mg/g, which is more than 12 times higher than the sorption capacity of activated carbon. The sorption capacity of montmorillonites samples of other compositions is somewhat lower, but they also have rather high values. The obtained values of the sorption capacity of synthetic montmorillonite samples correlate with the previously obtained results of the study of the sorption capacity of raw clays [2,5] and even slightly exceed them, which is due to the absence of impurity phases in the samples under study.

For samples of the kaolinite subgroup with different particle morphologies, the sorption capacity for albumin is significantly lower than that for montmorillonite, and is at the level of 60–80 mg/g for samples with platy, spherical, and tubular morphologies. For samples of kaolinite with a spherical particle morphology, the sorption capacity for albumin is even lower and is at the level of 25–40 mg/g. The sorption capacity of Beta zeolite is 53 mg/g. The sorption capacity for the albumin of all studied aluminosilicate samples exceeds the sorption capacity of activated carbon.

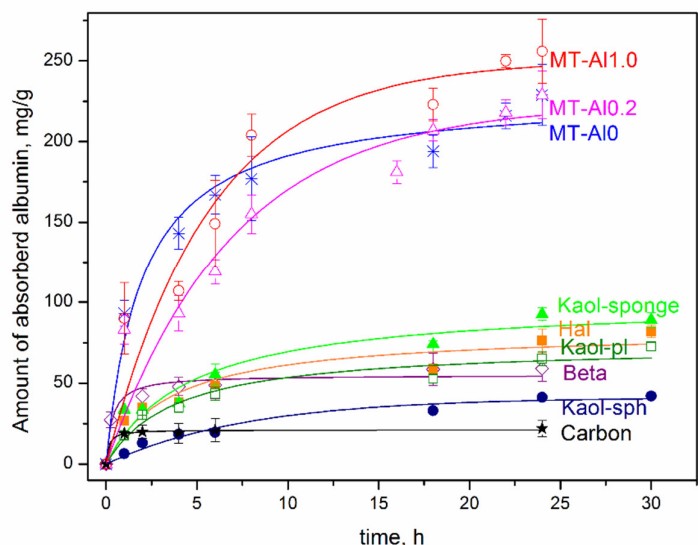

**Figure 5.** Kinetic curve of albumin by the aluminosilicate samples and activated carbon. Samples are designated in accordance with the designations presented in Table 1.

Such adsorption by aluminosilicate samples is associated with the features of their structure and surface properties. No direct relationship between the specific surface area of the samples and their sorption capacity for albumin was found. Thus, the highest values of the specific surface area are typical for MT-Al0 samples (549 m²/g), kaolinite nano-sponges (470 m²/g), and Beta zeolite (676 m²/g). However, the sorption capacity of these samples is not the highest. Montmorillonite samples are characterized by the highest sorption capacity for albumin, despite the highest values of the negative zeta potential of the surface. The high sorption capacity of montmorillonite compared to other samples is most likely due to its ability to increase the interlayer distance over a wide range (from 1 Å to complete exfoliation into individual layers) in the process of the adsorption of organic and inorganic molecules and their intercalation in the interlayer space [55,56]. As a result, albumin is located both on the outer (chips, edges, and outer surface of layers) and inner surfaces (interlayer space) of montmorillonite particles. The structures of other aluminosilicate samples do not have this feature, and albumin adsorption mainly occurs on the outer surface of the particles. On the other hand, the ability to intercalate and increase the interlayer distance leads to an increase in the adsorption equilibrium time for samples with the montmorillonite structure.

Based on the results of the graphical processing of the experimental data (Table 3), it was found that the sorption kinetics of all samples, except for MT-Al0.2 and MT-Al1.0 samples, is well described by a pseudo-second-order (PSO) equation: the theoretically calculated values of the sorption capacity $q_{calc}$ are close to those found experimentally, and the high approximation coefficient is 0.92–0.98. The PSO kinetic model is usually associated with the situation where the rate of the direct adsorption/desorption process is rate limiting. Within the framework of kinetic models, the rate constants of the process were calculated (Table 3). The PSO rate constants are the highest for the activated carbon and Beta zeolite samples, which is consistent with the short equilibration time in the system (1 and 2 h, respectively). The most adequate model for the MT-Al0.2 and MT-Al1.0 samples, taking into account the $q_{calc}$ values, is PFO.

**Table 3.** Parameters of the kinetic models of sorption of albumin on aluminosilicates with different morphologies.

| Samples Morphology | $q_{exp}$, mg/g | PFO Model | | | PSO Model | | |
|---|---|---|---|---|---|---|---|
| | | $q_{calc}$ | $k_1$ | $R^2$ | $q_{calc}$ | $k_2$ | $R^2$ |
| MT-Al0 | 229 ± 19 | 207 ± 11 | 0.32 ± 0.07 | 0.93 | 229 ± 10 | $(3 ± 1) 10^{-3}$ | 0.97 |
| MT-Al0.2 | 229 ± 15 | 222 ± 18 | 0.14 ± 0.03 | 0.92 | 278 ± 32 | $(5 ± 2)·10^{-4}$ | 0.93 |
| MT-Al1.0 | 256 ± 24 | 250 ± 18 | 0.17 ± 0.03 | 0.92 | 301 ± 30 | $(7 ± 3)·10^{-4}$ | 0.93 |
| Kaol-sph | 42 ± 4 | 41 ± 2 | 0.13 ± 0.02 | 0.96 | 52 ± 4 | $(2 ± 1) 10^{-3}$ | 0.98 |
| Kaol-sponge | 92 ± 2 | 86 ± 6 | 0.18 ± 0.04 | 0.89 | 101 ± 9 | $(2 ± 1) 10^{-3}$ | 0.93 |
| Kaol-pl | 72 ± 3 | 64 ± 4 | 0.22 ± 0.04 | 0.92 | 74 ± 5 | $(3.5 ± 1) 10^{-3}$ | 0.95 |
| Hal | 82 ± 4 | 72 ± 6 | 0.23 ± 0.06 | 0.87 | 83 ± 7 | $(3.4 ± 1) 10^{-3}$ | 0.92 |
| Beta | 59 ± 7 | 51 ± 3 | 2.9 ± 1.1 | 0.90 | 55 ± 3 | $(5 ± 0.2) 10^{-2}$ | 0.95 |
| Carbon | 22 ± 5 | 20 ± 1 | 2.5 ± 0.8 | 0.98 | 21 ± 1 | 0.34 ± 0.18 | 0.99 |

Figure 6 shows albumin adsorption isotherms by synthetic aluminosilicates, as well as by activated carbon. The symbols represent the experimental data, and the lines represent the model that best fits the data. Taking into account the high values of the correlation coefficients ($R^2$) and the close values of the experimental and calculated sorption capacity (Table 4), among the three nonlinear models, the Langmuir isotherm best describes adsorption on all samples, except for Hal and Kaol-pl. This model describes a homogeneous monomolecular adsorption process and assumes that the surface of a solid body contains a finite number of active centers with equal energy. For the Hal and Kaol-pl samples, the Freundlich equation is the most appropriate. According to the Freundlich model, the surface of the studied sorbents contains active centers with different affinity energies for adsorbate molecules. The value of 1/n can be considered as an indicator of the inhomogeneity of sorption centers: as the inhomogeneity increases, 1/n→0, and as the homogeneity of centers increases, 1/n→1. At the same time, the data obtained make it possible to characterize aluminosilicates as materials with a high concentration of sorption centers with different degrees of activity, which is consistent with the results of studying the distribution of active centers on the sample surface (Figure 4). The constant $K_F$ has a linear dependence on the adsorption capacity of the adsorbent, i.e., the larger this constant, the greater the adsorption capacity.

Table 2 presents the results of determining the sorption capacity of samples in relation to potassium and sodium cations, as well as vitamin B1 in the SBF medium. The results of the study of the hemolytic activity of the samples are also given there.

Based on the obtained results, it can be concluded that all studied samples absorb sodium and potassium cations from SBF in small amounts, potentially not leading to serious pathological changes. At the same time, two samples—MT-Al0 and kaolinite with nanosponge morphology—do not have a sorption capacity for these cations. The sorption capacity of aluminosilicates with respect to vitamin B1 is relatively high and is the highest for samples with a montmorillonite structure—MT-Al0.2, MT-Al1.0, Sap, and for Beta zeolite. The samples of the kaolinite subgroup with spherical, platy, and tubular particle morphologies are characterized by the lowest sorption capacity. The results of the vitamin B1 adsorption study generally correlate with the results of the BSA adsorption study and can be explained by the structural features of the studied samples.

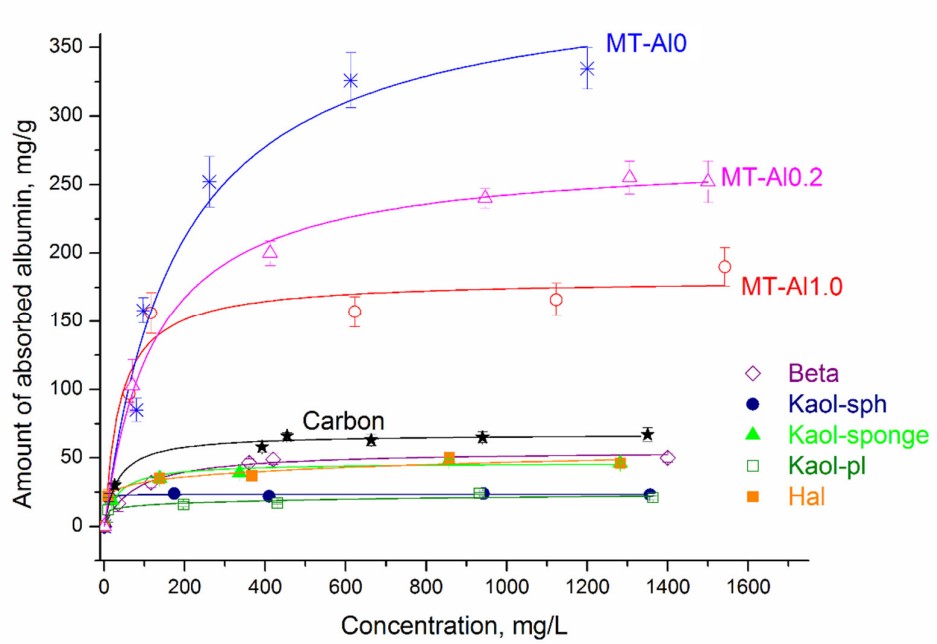

**Figure 6.** Langmuir and Freundlich adsorption isotherm plots. Langmuir isotherm: MT-Al0; MT-Al1.0; MT-Al0.2; Beta; Kaol-sph; Kaol-sponge; carbon samples. Freundlich isotherm: Hal and Kaol-pl.

**Table 4.** Equation constants of albumin sorption isotherms.

| Sample Morphologies | $q_{exp}$ | Langmuir Equation | | | Freundlich Equation | | | Temkin Equation | | |
|---|---|---|---|---|---|---|---|---|---|---|
| | | $q_m$ | $K_L$ | $R^2$ | n | $K_F$ | $R^2$ | $B_T$ | $A_T$ | $R^2$ |
| MT-Al0 | 334 ± 20 | 404 ± 34 | $(5 ± 1) 10^{-3}$ | 0.97 | 2.9 ± 0.7 | 30.5 ± 2.9 | 0.90 | 89.2 ± 7.5 | $(1 ± 0.1) 10^{-2}$ | 0.96 |
| MT-Al0.2 | 255 ± 12 | 272 ± 5 | $(8 ± 1) 10^{-3}$ | 0.99 | 364 ± 0.4 | 36.0 ± 8.0 | 0.96 | 50.0 ± 2.0 | $(11 ± 2) 10^{-2}$ | 0.94 |
| MT-Al1.0 | 190 ± 14 | 180 ± 11 | $(25 ± 9) 10^{-3}$ | 0.94 | 7.3 ± 2.5 | 67.0 ± 6.5 | 0.93 | 20.5 ± 6.3 | 4.6 ± 1.1 | 0.92 |
| Kaol-sph | 24 ± 1 | 23 ± 1 | 0.8 ± 0.3 | 0.99 | 49.2 ± 5.0 | 20.4 ± 1.4 | 0.97 | 1.0 ± 0.4 | 3.1 ± 0.2 | 0.97 |
| Kaol-sponge | 48 ± 1 | 46 ± 2 | $(8 ± 3) 10^{-3}$ | 0.98 | 5.2 ± 0.8 | 12.6 ± 2.3 | 0.97 | 6.8 ± 0.6 | 1.0 ± 0.5 | 0.96 |
| Kaol-pl | 24 ± 1 | 20 ± 2 | $(13 ± 1) 10^{-2}$ | 0.87 | 7.2 ± 2.0 | 8.1 ± 2.0 | 0.95 | 9.7 ± 1.6 | 2.2 ± 0.9 | 0.95 |
| Hal | 48 ± 3 | 43 ± 3 | $(8 ± 4) 10^{-2}$ | 0.91 | 6.2 ± 1.0 | 15.3 ± 2.6 | 0.98 | 5.4 ± 0.8 | 5.1 ± 0.3 | 0.97 |
| Beta | 50 ± 6 | 55 ± 2 | $(13 ± 2) 10^{-2}$ | 0.99 | 4.4 ± 1.2 | 18.0 ± 4.6 | 0.90 | 9.7 ± 1.6 | 2.2 ± 1.5 | 0.95 |
| Carbon | 63 ± 8 | 66 ± 2 | $(31 ± 7) 10^{-3}$ | 0.99 | 4.2 ± 0.8 | 14.3 ± 3.5 | 0.98 | 10.9 ± 1.2 | 0.6 ± 0.4 | 0.94 |

$q_m$—maximum sorption capacity (mg/g); $q_{exp}$—experimental value of sorption capacity (mg/g); Langmuir constant related to adsorption free energy (L/mg); $B_T$—constant related to the heat of adsorption (L/g); $K_F$—Freundlich constant related to adsorbent capacity (L/g); $A_T$—dimensionless Temkin isotherm constant.

Blood plasma is an aqueous solution of electrolyte, nutrients, metabolites, proteins, vitamins, trace elements, and signaling substances. The most important characteristic of a selective hemosorbent is the presence of sorption capacity in relation to pathogens, and its absence in relation to other blood components, particularly vitamins and microelements. In this case, the optimal hemosorbent should also not have the ability to destroy blood cells, that is, it should not have hemolytic activity. The results of the study of hemolytic activity, presented in Table 2, indicate that the greatest increase in hemolytic activity (toxicity) at a sample concentration of 10 mg/mL occurs in the series Sap<Carbon<Beta<Kaol-sph<Kaol-sponge<MT-Al0.2<Kaol-pl<MT-Al1.0<Hal. The presence and difference of the hemolytic activity in samples may be associated with differences in their chemical composition, surface properties, and particle shape. Thus, the dependence of the

hemolytic activity and cytotoxicity of aluminosilicates of the kaolinite subgroup on the morphology of their particles was shown earlier [57]. It was found that among single-phase samples with the same chemical composition $Al_2Si_2O_5(OH)_2$, samples with a tubular morphology have the highest toxicity, and samples with spherical particles have the lowest toxicity. In addition to this effect, the effect of the influence of the chemical composition on the hemolytic activity was found in the present work. Among the samples with the montmorillonite structure, samples with the highest aluminum content have the highest hemolytic activity. The sample of magnesium silicate montmorillonite $(Mg_3Si_4O_{10}(OH)_2 \cdot nH_2O)$ has the lowest hemolytic activity among all the studied samples, including activated carbon.

It should be noted that samples are considered non-toxic if their hemolytic activity does not exceed 5% [58]. With a decrease in the concentration of the studied samples to 0.3 mg/mL, the hemolytic activity of all samples decreases, and for most samples, reaches values not exceeding 5% (with the exception of samples MT-Al1.0 and Hal).

## 4. Conclusions

This work studied the possibility of using porous aluminosilicates with different structures and particle morphologies as hemosorbents. The sorption capacity of the samples in relation to the model medium molecular weight toxicants (BSA), vitamin B1, and alkaline cations in a simulated body fluid, as well as their hemolytic activity, were studied. It was established that the sorption capacity of aluminosilicate samples is largely determined by their structural features, porous textural characteristics and surface properties (charge and distribution of active centers on the surface). Thus, samples with a montmorillonite structure, which have the ability to increase the interlayer space over a wide range, have the highest sorption capacity with respect to BSA. However, the time to reach sorption equilibrium for such samples is quite long and amounts to approximately 24 h, which is also related to the peculiarities of their structure. The sorption capacity of zeolite samples is several times lower than montmorillonite, however, the sorption equilibrium is reached in 1 h. The Langmuir isotherm best describes adsorption on all samples, except for samples with nanotubular and platy particle morphology. For these samples, the Freundlich equation is the most appropriate. The hemolytic ability of the samples is largely determined by the morphology of the particles and the chemical composition of the samples. Thus, aluminosilicates with tubular particles and samples with the highest aluminum content have the highest hemolytic activity.

The studies of adsorption features and properties of porous aluminosilicates have shown that aluminosilicate sorbents can be considered potential hemosorbents. They have a high sorption capacity for medium molecular weight pathogens (up to 12 times that of activated charcoal for some samples) and low toxicity to blood cells. Directed hydrothermal synthesis makes it possible to obtain aluminosilicates with a given chemical and phase composition, certain surface properties, and porous textural characteristics. It is shown that the chemical composition, surface charge, particle morphology, and structural features determine the adsorption capacity and biological activity of the samples.

Based on the performed study, it can be concluded that the optimal option for the further development of new selective and non-toxic hemosorbents is synthetic magnesium silicate montmorillonite $(Mg_3Si_4O_{10}(OH)_2 \cdot nH_2O)$, since it has a significant sorption capacity with respect to BSA, modeling pathogens with an average molecular weight, lack of sorption capacity for potassium and sodium cations from the blood plasma medium, and low hemolytic activity.

**Author Contributions:** Conceptualization, O.Y.G.; Methodology, O.Y.G.; Validation, O.Y.G., Y.A.A., and E.Y.B.; Formal Analysis, O.Y.G., E.Y.B., N.M.V., and Y.A.A.; Investigation, Y.A.A., E.Y.B., and N.M.V.; Resources, O.Y.G.; Data Curation, O.Y.G., Y.A.A., N.M.V., and E.Y.B.; Writing—Original Draft Preparation, O.Y.G. All authors have read and agreed to the published version of the manuscript.

**Funding:** This study was funded by the Russian Science Foundation (project No. 22-23-00227, accessed on 21 December 2021, https://rscf.ru/project/22-23-00227/).

**Data Availability Statement:** Not applicable.

**Acknowledgments:** The authors are grateful to the employees of the Institute of Experimental Medicine, namely O.V. Shamova and E.V. Vladimirova, for conducting studies on the hemolytic activity of the samples. The authors are grateful to the administration of the St. Petersburg Technological Institute (Technical University) for the opportunity to use the equipment of the Engineering Center (X-ray diffractometer).

**Conflicts of Interest:** The authors declare no conflict of interest.

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
