# Peer review of "Adsorption Properties and Hemolytic Activity of Porous Aluminosilicates in a Simulated Body Fluid"

_2305-7084, doi:10.3390/chemengineering6050078_

Round 1

Reviewer 1 Report

Dear Authors

You did a good work and your paper can be accepted after making the following comments. The topic of the paper is attractive and interesting as the paper contains excellent and fantastic method and sequence. The topic is original and novel, and its good research done by the authors as it contains detailed information with simple way that can easily be understood. 

Abstract

1- The abstract do not contains the problem or the research problem. Why do you want to adsorb "bovine serum albumin (BSA), sodium and potassium 8 cations, and vitamin B1" are they dangerous or what. 

2- The main and brief results of the research work are not presented in the abstract.

Introduction

The introduction contains some previous researches or information related to the present work from general to specific while the last paragraph only introduce the research problem and solution and finally the most important results briefly no need to add much information here. Kindly adjust your introduction according to what is mentioned here. 

Results and discussion

1- SEM analysis results in different particles morphology and the authors must mentioned more details related to effect of particle morphology on the required adsorption process and which particle morphology is the best one.

Conclusion

The conclusion is so short with no concluded results or research problem and it needs to rewritten again. 

References

Recent references are required. 

Author Response

The authors would like to thank the Reviewer for valuable remarks and comments, which the authors have tried to take into account or comment on.

Point 1. The abstract do not contains the problem or the research problem. Why do you want to adsorb "bovine serum albumin (BSA), sodium and potassium 8 cations, and vitamin B1" are they dangerous or what. 

Response 1. Thank you very much for this comment. Information has been added to the abstract.

Point 2. The main and brief results of the research work are not presented in the abstract.

Response 2. The description of the main results has been added to the abstract.

Point 3. The introduction contains some previous researches or information related to the present work from general to specific while the last paragraph only introduce the research problem and solution and finally the most important results briefly no need to add much information here. Kindly adjust your introduction according to what is mentioned here.

Response 3. The authors tried to take this remark into account and corrected the introduction.

Point 4. 1- SEM analysis results in different particles morphology and the authors must mentioned more details related to effect of particle morphology on the required adsorption process and which particle morphology is the best one.

Response 4. The particle morphology does not make a decisive contribution to the sorption capacity of aluminosilicates. The sorption capacity is determined by the porous-textural characteristics, the distribution of active centers on the surface, the surface charge, and the chemical composition. The morphology of the particles largely determines the presence of toxicity of aluminosilicate particles. The article shows that among single-phase samples with the same chemical composition Al2Si2O5(OH)2, samples with tubular morphology have the highest toxicity, and samples with spherical particles have the lowest toxicity. In addition to this effect, the effect of the influence of the chemical composition on the hemolytic activity was found in the present work.

Point 5. The conclusion is so short with no concluded results or research problem and it needs to rewritten again.

Response 5. Thank you very much for the note. The authors expanded the conclusion.

Point 6. Recent references are required. 

Response 6. Several recent references have been added.

Reviewer 2 Report

The research is of interest and the manuscript is well constructed and readable. The results are well presented too.

However, improvements are required to corret minor errors, clarify some concepts, add missing data, and support some of the conclusions.

MINOR:

- Line 49: please, better specify what do you mean with “sufficient number of hemosorbents”;

- Line 52: typographic error “...certain methods. sterilization without…”;

- Line 54: “...according to some researchers…”, please add the related references;

- Line 55: “...inferior to other adsorbents…” these other adsorbents should be specified;

- Line 79: please, provide the reference for BSA molecular size, and how it was determined.

- Lines 107 and 406: “...montmorillonites have the ability to change the interlayer distance from 10 to 100 nm or more...”, for sure the authors means from 10 to 100 Å, not nanometers; furthermore, specific references should be added (e.g., Lepoitevin et al. 2014 Applied Clay Science 95, 396-402 reports interlayer distances of tens of Å). References 47 and 48 (even if good generic references for montmorillonite intercalation studies) do not support such values, with just few Å shift for the basal reflection.

- Line 160: please add the time/step (integration time) of the XRD acquisition.

- Lines 204-205: the sentence is not complete. Please, check.

- Lines 243-244: you introduce two concentration values (10 mg/mL and 0.1 mg/mL) without explaining why you are considering two different concentrations, and why you selected these two values. Furthermore, they are not in line with the concentrations reported in Table 2 ( 10 mg/mL and 0.3 mg/mL).

- Line 253: please, consider using “hydrous aluminosilicates” instead of “aqueous aluminosilicates”.

- Lines 269-271: please check, “Fig.2f” should be “Fig.2d”; “Fig.2e” is not cited; “Fig.2e” should be “Fig.2h”.

- Lines 326 and 328: check if pKa 4.1 is the correct value.

- Line 408: please, better define and specify what you consider “inner surfaces of montmorillonite particles”.

- Line 475: consider changing “it should have hemolytic activity” with “it should not have hemolytic activity”.

MAJOR:

- Abstract: no indications about the results of the research work are cited, except the generic last two sentences (lines 21,22) that should be briefly expanded to cite at least the most relevant specific result of the work;

- Line 138: “Samples corresponding to the Al2Si2O5(OH)4 kaolinite formula were synthesized under conditions that made it possible to obtain various particle morphologies — spherical, sponge and platy”. These conditions should be specified and described.

- Lines 172-175: no data are provided about the acquisition parameters of the SEM measurements: vacuum level, acceleration voltage, current, detected signals. Please, add these fundamental data.

- Line 262-271: some data are missing about dimensions of synthesized minerals: please provide at least the average size of saponite and montmorillonite particles, kaolinite particles with nanosponge morphology, average lateral size of kaolinite platy particles.

- Line 278 “similar shapes of hysteresis loops” and line 280 “The hysteresis loops are of different shapes” appears to be contradictory. Please clarify.

- Line 410 (and 405): “...the ability to intercalate and increase the interlayer distance leads to…”, “...due to its ability to increase the interlayer distance...”.The authors claim intercalation in the interlayer space as cause of (1) the high sorption capacity of montmorillonite, and (2) an increase in the adsorption equilibrium time for samples with the montmorillonite structure. This hypothesis should be verified by XRD analysis of the change of the interlayer distance of montmorillonite (in air, room temperature, humidity and pressure) before interaction, after interaction with the solution without BSA and with BSA.

- Finally, can you comment about the relatively high sorption capacity of aluminosilicates with respect to vitamin B1, being the absence of vitamins sorption capacity one of the most important characteristic of a selective hemosorbent?

Author Response

The authors would like to thank the Reviewer for a careful reading of the manuscript and for valuable comments and remarks, which the authors tried to take into account and make the appropriate changes and additions.

Minor

Point 1. Line 49: please, better specify what do you mean with “sufficient number of hemosorbents”;

Response 1. Thank you very much for this comment. Corresponding corrections have been made to the text of the manuscript. The word “sufficient” has been changed to “significant”

 Point 2. Line 52: typographic error “...certain methods. sterilization without…”;

Response 2. Corrections have been made to the text.

Point 3. Line 54: “...according to some researchers…”, please add the related references;

Response 3. Relevant references have been added.

Point 4. Line 55: “...inferior to other adsorbents…” these other adsorbents should be specified;

Response 4. Adsorbents are specified.

Point 5.  Line 79: please, provide the reference for BSA molecular size, and how it was determined.

Response 5. Relevant references have been added.

 Point 6. Lines 107 and 406: “...montmorillonites have the ability to change the interlayer distance from 10 to 100 nm or more...”, for sure the authors means from 10 to 100 Å, not nanometers; furthermore, specific references should be added (e.g., Lepoitevin et al. 2014 Applied Clay Science 95, 396-402 reports interlayer distances of tens of Å). References 47 and 48 (even if good generic references for montmorillonite intercalation studies) do not support such values, with just few Å shift for the basal reflection.

Response 6. Thank you very much for this comment. Of course, the authors meant angstroms, not nanometers. Corresponding changes have been made to the text. The phrase has been corrected as follows “montmorillonites have the ability to change the interlayer distance over a wide range - from 1 Å to complete exfoliation into individual layers”.

Point 7.  Line 160: please add the time/step (integration time) of the XRD acquisition

Response 7. Information has been added.

 Point 8. Lines 204-205: the sentence is not complete. Please, check.

Response 8. Thank you very much for the note. Information has been added.

Point 9.  Lines 243-244: you introduce two concentration values (10 mg/mL and 0.1 mg/mL) without explaining why you are considering two different concentrations, and why you selected these two values. Furthermore, they are not in line with the concentrations reported in Table 2 ( 10 mg/mL and 0.3 mg/mL).

Response 9. Concentrations in the range of 0.1 to 5 mg/ml are most often used in the study of hemolytic activity of drugs, as well as other compounds, such as crystalline silicon (0.5, 1, 2, and 5 mg/ml.) (W. Hadnagy, B. Marsetz, H. Idel. Hemolytic activity of crystalline silica – Separated erythrocytes versus whole blood. International Journal of Hygiene and Environmental Health. V. 206, Issue 2, 2003, P. 103-107). More specifically, the choice of concentrations is determined both by the nature of the object under study and by the tasks of its use. There are very few studies of the hemolytic activity of aluminosilicates in the literature. Our studies of hemolytic activity (O.Y. Golubeva, Y.A. Alikina, E.Y. Brazovskaya, Particles Morphology Impact on Cytotoxicity, Hemolytic Activity and Sorption  Properties of Porous Aluminosilicates of Kaolinite Group., Nanomater. (Basel, Switzerland). 12 (2022). https://doi.org/10.3390/nano12152559) have shown that most synthetic aluminosilicates have a rather low hemolytic activity in the region of low concentrations (up to 5 mg/ml). For some particle morphologies (nanotubes) at higher concentrations (10 mg/ml), hemolytic activity becomes significant. Two concentrations were chosen for this manuscript, 0.3 and 10 mg/ml, to show the possible dependence of hemolytic activity (or lack of it) on the sample concentration. This may be important in the case of using the material as a hemosorbent, where the concentration of the sorbent during its interaction with blood will be high.

 Point 10. Line 253: please, consider using “hydrous aluminosilicates” instead of “aqueous aluminosilicates”.

Response 10. Сhanges made to the text.

Point 11. Lines 269-271: please check, “Fig.2f” should be “Fig.2d”; “Fig.2e” is not cited; “Fig.2e” should be “Fig.2h”.

Response 11. Thank you very much for your comment. Appropriate corrections have been made to the text.

 Point 12. Lines 326 and 328: check if pKa 4.1 is the correct value.

Response 12. Should be -4.4. Corresponding changes have been made to the text.

Point 13. Line 408: please, better define and specify what you consider “inner surfaces of montmorillonite particles”.

Response 13. Relevant clarifications have been added to the text of the manuscript.

Point 14. Line 475: consider changing “it should have hemolytic activity” with “it should not have hemolytic activity”.

Response 14. Changes have been made to the text.

MAJOR:

Point 15. Abstract: no indications about the results of the research work are cited, except the generic last two sentences (lines 21,22) that should be briefly expanded to cite at least the most relevant specific result of the work;

Response 15. Additional information about the results obtained has been added to the abstract.

 Point 16.  Line 138: “Samples corresponding to the Al2Si2O5(OH)4 kaolinite formula were synthesized under conditions that made it possible to obtain various particle morphologies — spherical, sponge and platy”. These conditions should be specified and described.

Response 16. The synthesis conditions (temperature, synthesis duration) are presented in Table 1. A more detailed description of the synthesis conditions is presented in our referenced articles (Synthesis of all samples was carried out under hydrothermal conditions according to previously developed methods [20,30–34]).

Point 17. Lines 172-175: no data are provided about the acquisition parameters of the SEM measurements: vacuum level, acceleration voltage, current, detected signals. Please, add these fundamental data.

Response 17. The information was added to the text.

Point 18. Line 262-271: some data are missing about dimensions of synthesized minerals: please provide at least the average size of saponite and montmorillonite particles, kaolinite particles with nanosponge morphology, average lateral size of kaolinite platy particles.

Response 18. The information was added to the text of the manuscript.

 Point 19. Line 278 “similar shapes of hysteresis loops” and line 280 “The hysteresis loops are of different shapes” appears to be contradictory. Please clarify.

Response 19. Corresponding corrections have been made to the text of the manuscript.

Point 20. Line 410 (and 405): “...the ability to intercalate and increase the interlayer distance leads to…”, “...due to its ability to increase the interlayer distance...”. The authors claim intercalation in the interlayer space as cause of (1) the high sorption capacity of montmorillonite, and (2) an increase in the adsorption equilibrium time for samples with the montmorillonite structure. This hypothesis should be verified by XRD analysis of the change of the interlayer distance of montmorillonite (in air, room temperature, humidity and pressure) before interaction, after interaction with the solution without BSA and with BSA.

Response 20. On the one hand, the conclusion that the high sorption capacity of montmorillonite and the increase in the time to reach equilibrium is associated with its ability to increase the interlayer distance is the assumption of the authors, and on the other hand, this ability of raw montmorillonite is a well-known fact repeatedly proven by various researchers using various compounds, including albumin. In particular, in the works of the authors on the study of the sorption capacity of synthetic montmorillonites with respect to methylene blue, vitamins, and conjugates of silver nanoparticles [(O.Y. Golubeva et al., 2018; O.Y. Golubeva & Pavlova, 2016; Olga Yu. Golubeva et al., 2015)] these data has been presented multiple times. Therefore, the authors considered it possible not to confirm this conclusion by additional studies of changes in the interlayer distance of montmorillonite samples during albumin adsorption.

References:

Golubeva, O.Y., Brazovskaya, E. Y., & Shamova, O. V. (2018). Biological activity and sorption ability of synthetic montmorillonite modified by silver/lysozyme nanoparticles. Applied Clay Science, 163. https://doi.org/10.1016/j.clay.2018.07.015

Golubeva, O.Y., & Pavlova, S. V. (2016). Adsorption of methylene blue from aqueous solutions by synthetic montmorillonites of different compositions. Glass Physics and Chemistry, 42(2). https://doi.org/10.1134/S1087659616020073

Golubeva, Olga Yu., Pavlova, S. V, & Yakovlev, A. V. (2015). Adsorption and in vitro release of vitamin B1 by synthetic nanoclays with montmorillonite structure. Applied Clay Science, 112113, 10–16. https://doi.org/https://doi.org/10.1016/j.clay.2015.04.013

Point 21. Finally, can you comment about the relatively high sorption capacity of aluminosilicates with respect to vitamin B1, being the absence of vitamins sorption capacity one of the most important characteristic of a selective hemosorbent?

Response 21: The lack of sorption capacity for vitamins and minerals is a desirable characteristic of a selective hemosorbent, but not the main one. The magnitude of the sorption capacity for vitamins should be considered in conjunction with the sorption capacity of the sorbent for pathogens and toxins, as well as other minerals, as well as its toxicity.

Round 2

Reviewer 2 Report

The authors answered satisfactorily to all the requests.

Please, check:

- line 277: " ... layers self-organized into larger micron size particles (Fig. 2 a-c). According to previous studies [33], the average particle size of montmorillonite is about 20 nm ...";

- line 167: "... resolution 0.0002 ) ...".

Author Response

Point 1. line 277: " ... layers self-organized into larger micron size particles (Fig. 2 a-c). According to previous studies [33], the average particle size of montmorillonite is about 20 nm ...";

Response 1. The authors have made corrections. the word “particles” is replaced by “agglomerates”.

Point 2. - line 167: "... resolution 0.0002 ) ...".

Response 2. Information has been removed.